# Effect of Transgene Location, Transcriptional Control Elements and Transgene Features in Armed Oncolytic Adenoviruses

**DOI:** 10.3390/cancers12041034

**Published:** 2020-04-23

**Authors:** Martí Farrera-Sal, Cristina Fillat, Ramon Alemany

**Affiliations:** 1VCN Biosciences S.L., 08174 Sant Cugat, Spain; 2ProCure and Oncobell Programs, Institut Català d’Oncologia/Bellbitge Biomedical Research Institute, 08908 Hospitalet de Llobregat, Spain; 3August Pi i Sunyer Biomedical Research Institute (IDIBAPS), Rare Diseases Networking Biomedical Research Center (CIBERER), University of Barcelona, 08036 Barcelona, Spain; CFILLAT@clinic.cat

**Keywords:** oncolytic adenoviruses (OAds), adenoviruses (Ads), cancer, transgene

## Abstract

Clinical results with oncolytic adenoviruses (OAds) used as antitumor monotherapies show limited efficacy. To increase OAd potency, transgenes have been inserted into their genome, a strategy known as “arming OAds”. Here, we review different parameters that affect the outcome of armed OAds. Recombinant adenovirus used in gene therapy and vaccination have been the basis for the design of armed OAds. Hence, early region 1 (E1) and early region 3 (E3) have been the most commonly used transgene insertion sites, along with partially or complete E3 deletions. Besides transgene location and orientation, transcriptional control elements, transgene function, either virocentric or immunocentric, and even the codons encoding it, greatly impact on transgene levels and virus fitness.

## 1. Introduction

In the 1950s–1970s, anecdotal reports of spontaneous tumor remissions as a result of naturally acquired viral infections prompted the clinical concept of using pathogenic viruses to cure cancer, known as virotherapy. One of the first clinical trials was done in 1956 using wild-type adenovirus (Ads) as a therapeutic agent in cervical cancer. It was demonstrated to be safe with mild flu-like symptoms as the main adverse effects [1], and apparent efficacy with marked tumor necrosis or liquefaction of tumors. However, the antitumor responses lasted only months and did not translate into an overall survival benefit, unlike the promising chemotherapeutic agents, at that time.

In 1996 Frank MacCormick proposed the use of a mutant adenovirus to treat p53-deficient tumors [2]. In the following years this virus, renamed as ONYX-015, was tested in numerous clinical trials with limited efficacy. A similar virus was developed in China under the name Oncorine. In 2005, it gained approval for the treatment of head and neck tumors by intratumoral injection in combination with systemic chemotherapy [3]. The response rate with the combination doubled that obtained with chemotherapy alone, but no overall survival improvements were reported.

To increase the efficacy of oncolytic adenoviruses major research efforts have been directed to enhance tumor-targeting by capsid modifications, and tumor-selective replication using promoters and gene deletions. These aspects have been broadly reviewed in the literature [4,5,6]. Replication-selective oncolytic adenoviruses (OAds) with these modifications, such as CV706, CG7870, AdD24RGD, and ICOVIR5, entered clinical trials. However, despite occasional striking tumor responses, clinical efficacy remained low [7,8,9,10,11]. Thus, the potency of oncolytic adenoviruses should be further increased to become a real and viable cancer therapy. The insertions of transgenes into their genome, also known as “arming OAds” [12], aims to the destruction of neighboring and distant uninfected cancer cells by the direct or indirect effect of the transgene-encoded therapeutic proteins or RNA.

Adenovirus has several advantages compared to other oncolytic viruses for transgene insertion. All RNA viruses except orthomyxoviruses (e.g., influenza virus) and retroviruses replicate in the cytoplasm and therefore nuclear transcriptional control elements, such as tumor selective promoters, are non-functional. In contrast, DNA viruses except poxviruses (e.g., vaccinia virus) replicate in the nucleus and are compatible with the use of eukaryotic promoters. Viruses with small compact genomes cannot accept transgenes (picornavirus, reovirus, and parvovirus). Besides, adenovirus stands out from the rest of transgene-carrying viruses (Vesicular Stomatitis Virus, Measles Virus, Newcastle disease Virus, Herpes Virus, and Vaccinia Virus) because it has a highly regulated temporal sequence of viral gene expression starting with E1a which allows tuning transgene expression.

Here we review the different strategies used to arm oncolytic adenoviruses. Although the nature of the therapeutic transgene would be the first important feature for an armed-OAd, it is crucial to note that the amount, and timing of transgene expression are pivotal for this arming strategy. In this review, we explore how transgene location, different transcriptional control elements, the transgene codon usage, and the transgene itself affect its expression, kinetics, and viral replication.

## 2. Transgene Location Affects Expression Levels and Fitness of Armed OAds

Adenoviridae is a family of icosahedral, non-enveloped viruses with an approximately 30–40 kb linear double-stranded DNA genome. In humans, more than 50 different serotypes associated in six groups (A to F) have been discovered. Nevertheless, the most common for oncolytic therapy are serotypes Ad2 and Ad5 from group C. These are the best characterized among all the serotypes; replicate to higher yields than most other human Ad serotypes, and DNA sequences and tools are readily available for their genetic modification [13].

Adenoviruses maximize their genome coding capacity by generating early and late transcription units (active before or after virus DNA replication, respectively) controlled by multiple promoters, which in turn generate different RNAs and proteins by alternative splicing [14,15] (Figure 1). The strategy to insert additional genes stems from the use of adenoviruses as gene transfer vectors in the fields of gene therapy and vaccination. The transcription units Early 1 (E1), E3, and E4 were used as insertion sites for exogenous DNA sequences in the generation of recombinant Ads [16,17,18,19]. E1 is needed for viral replication and its defect is complemented by E1-expressing vector-packaging cell lines, such as HEK 293. In contrast, E3 mostly encodes immune-inhibitory proteins non-essential for viral replication in vitro [16,20,21]. An advantage of using the E3 region to insert the transgene is that it provides genomic space. Recombinant adenoviruses will only stably accommodate approximately 2 kb of additional DNA beyond the size of the normal genome [22]. Thus, deletions are needed to encode transgenes larger than 2kb and E1 and E3 deleted adenoviruses were used in the gene therapy field as a backbone for large transgene insertions.

In contrast to gene therapy vectors, OAds replicate in cancer cells. Thus E1 cannot be deleted entirely. The partial deletion of E1b55K was the first deletion postulated to be dispensable in tumor cells, based on the rationale that the p53-blocking function of this protein is not needed in p53-deficient cells [2]. Freytag et al. described the first armed-OAd with an E1b55K deletion and a transgene cassette with a cytomegalovirus promoter (CMVp) controlling a *cytosine deaminase* gene (CD) fused to the herpes virus thymidine kinase (TK) (Figure 1). This virus exhibited tumor cell specificity and enhanced viral therapy thanks to transgenes [23,24]. Similarly, the *thymidine kinase* (TK) gene was inserted in an E1b55K and E3-deleted adenovirus after the E1 region [25]. Since then, entire or partial E3 deletions have been used to generate genomic space to insert expression cassettes formed by an exogenous promoter, the transgene and polyadenylation sequences in different locations such as E1, E3 or E4 (Figure 1).

To bypass the need of exogenous promoters in OAds the group led by Terry Hermiston at Onyx substituted E3 genes with transgenes maintaining the endogenous promoters and polyadenylation sites, and endowing the transgene with the expression kinetics of the substituted *E3* gene [26,27,28]. They replaced the E3 6.7K and gp19K open reading frames (ORFs, Figure 1) with cytosine deaminase (CD) or tumor necrosis factor alpha (TNFα) resulting in high levels of transgene expression and viral replication dependency [26]. However, as gp19K is responsible for the retention of MHC-I inside the cell, these viruses are more susceptible to be eliminated by cytotoxic T-lymphocytes (CTLs). They also substituted the E3 gp11.6K adenovirus death protein (ADP) for CD or TNFα. As ADP is involved in the release of viral progeny, these viruses presented a delayed cytopathic effect that could be useful for an extended transgene production period, where the infected cell operates as a “factory” [27]. In contrast to this lytic delay to enhance transgene production, Rohmer et al. proposed the opposite strategy combining accelerated tumor cells lysis and transgene expression. The deletion of the anti-apoptotic E1b19K gene substantially increases the adenoviral cell killing [29,30]. The enhancement of apoptosis-dependent early viral release correlated with an increased transgene expression [31]. Hence, other Ad5 mutants with early viral release/enhanced spread phenotype in tumors could be considered to increase the transgene expression [32,33,34].

The Hermiston group also published the replacement of E3B adenoviral genes (RIDα/β and 14.7K, Figure 1) with TNFα. They obtained the highest levels of transgene compared with the other two insertion sites (6.7K/19K or ADP). This site conferred late gene kinetics and did not interfere with viral cytopathic effect [28]. These E3 replacements could be used simultaneously to obtain a multi-therapeutic gene expression with native viral promoters [35].

Later on, in 2005, the same group developed transposon-based approaches to scan an E3-deleted adenoviral genome for new expression cassette insertion sites. Four different locations were described: within the E1A promoter, within the E1B gene, between E1A and E1B, and within the E4 untranslated region (Figure 1) [36]. A similar approach was done with a transgene cassette controlled by different splice acceptor (SA) variants. They found viable viruses with insertions before 14.7K of E3, between E3 and L5, after L5, and between E4 and ITR (Figure 1). Curiously, most of the inserts were in rightward orientation (left to right on Ad5 genome) [37]. The level of transgene expression and viral replication of resulting armed-OAds depended on transgene, promoter, and cassette orientation. Therefore an optimal insertion site cannot be universally defined.

The impact of transgene orientation had been highlighted previously in gene therapy vectors. Foreign genes inserted rightwards within E1A, regardless of the promoter, expressed higher levels than leftward [38]. Transgenes encoded under exogenous promoters in E3 were efficiently transcribed in both orientations [39]. But others found that DNA inserted rightward in E3 was expressed at higher levels than leftward orientation [40,41,42]. In these cases, the inserted gene lacked a strong exogenous promoter, so the expression was mostly due to transcription initiating upstream to transgene by E3 promoter or Major Late promoter (MLP). Hence, transgene orientation should be considered at the time of cloning. In the absence of an exogenous promoter the orientation should coincide with the replaced adenovirus gene.

Based on the mentioned strategies, most OAds have major deletions within the E3 region. Notably, E3-deleted viruses were reported to be cleared much more rapidly than wild-type viruses and presented lower activity in immunocompetent in vivo models [43,44,45], therefore the E3 deletion may contribute to the fast clearance of adenoviruses in patients. To circumvent this, different strategies were designed to insert transgenes in a complete Ad backbone, despite that Ad5 packaging is limited to 38 kb (2 kb over the wild type size).

For non-E3-deleted OAds the first reported insertion site was right downstream the *fiber* gene (fiber is encoded in the late transcription unit 5 -L5-, Figure 1), previously defined also in E3-deleted backbones [37]. When expressed from the major late promoter as part of the long primary pre-mRNA transcript, the inserted transgene in this position forms a new transcription unit also known as a Late 6 (L6) unit. A splice acceptor before the transgene start codon allows the formation of the correct mRNA for the transgene. As a first step to test the feasibility of adding a L6 into Ad5, a GFP [46] or cytosine deaminase [47] were inserted with splice acceptors. Transgene expression was dependent on viral replication without any fitness loss. Alternatively, new insertion sites in the late transcription unit 3 (L3) were described cloning the *GM-CSF* gene between the hexon gene and the 23K protease gene or downstream the 23K under a splice acceptor (Figure 1). After the hexon significantly affected viral replication, but downstream the *23K* gene showed the highest levels of transgene expression when compared to E3-19K or E3-14.7K replacements. Linking transgene expression to the L3 transcripts, which are abundant at the late phase, achieved high levels of transgene expression without affecting viral growth in a non-deleted adenoviral backbone [48].

## 3. Genetic Elements that Control Transgene Expression in the Context of Armed OAds

Apart from the insertion site, the transcriptional control of the therapeutic transgene is one of the main essential features when designing an OAd. With regard to the transcriptional control of the transgene there have been two main approaches in the field. First, transgenes are inserted as autonomous expression cassettes containing an exogenous promoter and a polyadenylation signal. In these cases, the transgene expression does not depend on virus replication. Alternatively, transgenes are inserted into adenoviral transcription units, taking advantage of the viral gene expression machinery by internal ribosome entry sites (IRES), splice acceptor sites (SA) or protein fusions using 2A self-cleaving peptide linkers [49]. This strategy exploits the viral mechanisms of transcription and mimics the timing of transgene expression within the viral replication cycle and, accordingly, transgenes linked to late phase genes depend on viral replication. This late-phase selective expression may be used to improve the safety profile compared with vectors armed with constitutive promoters or early adenoviral promoters.

Alternatively, microRNAs (miRNA) can be used to regulate transgene expression. They are short-length non-coding RNAs that bind to complementary target sequences, leading to suppression of gene expression via post-transcriptional regulation. Their use to restrict OAds replication has been reviewed [50]. Moreover, miRNAs have been used to control transgene expression in Ad vectors [51,52,53], and therefore they are an additional strategy to regulate transgene expression in OAds.

### 3.1. Exogenous Promoters

Exogenous promoters have been extensively used in adenovirus vectors for gene therapy and they were used in the first armed OAds. The most widely used promoter is the human cytomegalovirus promoter (CMVp), a constitutive promoter trans-activated by early adenoviral proteins but easily silenced in mammalian systems. CMVp confers high levels of transgene transcription at early timepoints independent of virus replication. Transgenes are transcribed in every infected cell, even in normal tissues. To achieve tumor-selective expression, transgene transcription has been controlled by cellular promoters overactive in tumor cells. For instance, an OAd encoding GFP controlled by a fusion promoter between human telomerase reverse transcriptase (hTERT) and a small fragment of the CMVp [54]. Progression elevated gene-3 (PEG-3) is selectively expressed in diverse cancer cells, but it has limited activity in normal tissues. Aiming to control also the timing of transgene expression, an OAd was armed with a CD-TK fusion gene under heat shock protein 70 (hsp70) promoter. This promoter allows expression only after induction of 41 ℃ during 1 h [55]. Nevertheless, hsp70 promoter efficiently controlled transgene expression by heat-shock induction at early stages of viral infection, but not after adenovirus genome replication, leading to uncontrolled expression at late stages [56].

The main limitation of exogenous promoters is that they require large DNA fragments, increasing the size of the transgene cassette, and thus compromising the packaging limit of OAds. Moreover, the transgene kinetics could not be carefully controlled and becomes independent of virus replication.

### 3.2. Internal Ribosome Entry Sites (IRES) and 2A Sequences

Internal ribosome entry sites (IRESs) are highly structured viral sequences that facilitate biscistronic gene expression by internal RNA translation initiation in addition to the cap-dependent translation at the 5′ of mRNAs. In consequence, IRES allow trangene expression with the same kinetics as the linked viral gene. When connected to a late gene, transgene expression can be restricted to the sites of virus replication, which is tumor-selective in the case of OAds. The IRES sequence from the encephalomyocarditis virus was used to express transgenes from early and late Ad transcription units, obtaining higher transgene levels than CMV-armed OAds at late stages [47,57,58,59,60]. A spacer sequence between the stop codon and IRES appeared to be crucial for optimal expression [57]. Although IRES preserve transgene control linked to virus replication; the large size of IRES (aprox. 600 bp) is an obstacle to insert long trangenes in non-deleted adenoviral backbones. Shorter IRES have been described (elF4G—339 bp), but they are still far from the size economy of 2A sequences and splice acceptors.

The viral 2A sequences are short peptide sequences that facilitate multiprotein expression from one open reading frame by a process termed ribosomal skipping [61,62]. Its use in OAds was first described using the 2A sequences from the foot-and-mouth disease virus F2A and the porcine teschovirus-1 P2A to co-express GFP with adenoviral pIX [63]. Later, transgene selectivity and expression levels were compared between IRES or thosea asigna virus T2A sequence linked to the fiber gene, or cloned after E4 by a SA. Higher expression was obtained with IRES or SA than T2A [64]. Most concerning, 2A sequences reduced the replication of the OAds. Thus, other 2A sequences should be tested, or as an alternative the use of the caspase-8 cleavage site, which has been used between transgenes of armed-OAds to generate bicistronic transcripts [65,66].

### 3.3. Splice Acceptors

Alternative splicing was first described for adenovirus late mRNAs [67] and it can be used to create new transcripts in the adenovirus genome. The short sequence of SA, usually less than 50 bps, is one of the main advantages for large insertions into the Ad genome.

The IIIa splicing unit and splice enhancer (3VDE) was the first one to be studied in detail [68]. We have used this sequence to express GFP, TK, hyaluronidase, and bispecific T-cell engagers as transgenes after the fiber, as a L6 unit [46,60,69,70,71]. Ad40 and Ad41 express two fibers simultaneously from two SA, thus exploring the SA from these fibers was highly attractive to arm Ad5 with transgenes after the fiber. The Ad41 long fiber SA was used to introduce the *cytosine deaminase* gene after Ad5 adenoviral fiber [47]. Alternatively, the SA from Ad40 long-fiber produced at least 50-fold higher luciferase compared to a CMV-driven cassette at late time points [72]. An artificial splice acceptor site derived from the *beta globulin* gene (branch point and splice acceptor sequence (BPSA) has been used in the adenoviral context to express transgenes [37]. It was also reported the Ad40SA as stronger SA than BPSA [73]. However, comparisons between these splice acceptors (IIIa, Ad40 SA, Ad41 SA, and BPSA) in terms of transgene expression levels and viral fitness are not available. In a head to head study comparing SA and IRES using CD [47] and Luciferase [64], the IRES resulted in the expression of more transgene but at the expense of selectivity. Thus, the IRES approach might be attractive when the insert size and transgene toxicity are not a limitation. Otherwise, SA could be a suitable alternative but the transgene sequence has to be analyzed for cryptic splice acceptors to prevent undesired or aberrant transgene products [64].

## 4. Transgene-Related Parameters that Influence the Outcome of Armed OAds

Several transgene features have to be considered when arming OAds. As mentioned above, the length of the transgene has a major impact on the design of armed OAds. The insertion of transgenes longer than 2kb requires deletions in the adenoviral backbone to be adequately packaged into virions. In these cases, saving genome space to avoid excessive adenoviral gene deletions is crucial, as we have seen with the detrimental in vivo impact of E3 deletions. Thus, embedding the transgene into deleted transcription units taking advantage of endogenous promoters or the use of short sequences such as SA is the best option to control transgene transcription.

The transgene sequence may impact on the arming outcome at different levels. Eukaryotic transgenes may present introns in their sequences that are important for post-transcriptional stability and nucleocytoplasmic transport. Indeed, an increase of transgene expression in vitro and in vivo was reported in gene therapy vectors using introns in the expression cassette [74]. However, as they increase the transgene length, introns have not been used in OAds.

The set of codons encoding the transgene, known as the codon usage, might also affect its expression and viral fitness. We reported that adenovirus proteins could be grouped according to their codon usage. In particular, most early viral regulatory proteins used codons with A/T ending, while proteins implicated in replication and virion formation, including structural proteins, used codons ended in G/C. However, the fiber presented a different codon usage compared to the rest of the structural proteins, which was also poorly adapted to the human codon usage. Optimization of the Ad5 *fiber* gene sequence to the human codon usage in an adenovirus decreased fiber expression and reduced expression of other structural proteins such as hexon and penton, compromising viral fitness. Furthermore, the introduction of GFP as a late gene after fiber under a splicing acceptor further reduced viral fitness [75].

According to these results, changing a given protein codon usage, such as the fiber, could generate an imbalance in aminoacylated-tRNA (aa-tRNA) use between structural proteins resulting in a competition for the same aa-tRNA host pool leading to decreased late protein expression and viral fitness. This could explain why the insertion of a transgene as a late gene, in this case GFP, with a codon usage competing with structural proteins further attenuated virus replication. The results suggest that transgene codon usage should be adapted to avoid competing with the aa-tRNA frequently used by late genes, a feature to consider when arming OAds.

The level of transgene needed for the therapeutic effect or the potential cytotoxicity of the transgene should be taken into account for the selection of the appropriate transgene location or transcriptional control elements. For instance, non-cytotoxic transgenes could be encoded in early regions with constitutive promoters or linked to adenoviral late genes. Otherwise, cytotoxic transgenes that may hamper virus-producer cells before completing the viral replication cycle should be encoded in late phases and at lower amounts to allow proper virus replication. The constitutive or immediately early phase expression does not restrict the transgene transcription to the tumor and all infected cells produce transgene, independently of virus replication. Regarding this, toxicity in vivo has to be addressed, especially in systemic administrations.

Therefore, transgene function is a key feature for arming design. The function of transgenes can be considered virocentric or immunocentric. Virocentric transgenes would enhance virus cytotoxicity, yields or spread. Immunocentric transgenes aim to enhance the immune responses elicited during the oncolysis. The rationale behind immunocentric arming is that anti-tumor immune response may be the ultimate mechanism of efficacy for OAds. Here, we review some examples of both types of transgenes and summarize the armed-OAds in clinics.

### 4.1. Virocentric Transgenes

Transgenes aiming to improve the oncolytic effect, enhance virus spread, or kill bystander non-infected cells can be considered virocentric. Initially, genes used to arm OAds were prodrug-converting enzymes for molecular chemotherapy, known as suicide genes. These enzymes promote the conversion of intravenously administered non-toxic prodrugs to toxic drugs in adenovirus-infected cells, resulting in high local concentrations in the tumor. They induce a bystander effect of the toxic drug with limited systemic toxicity. Some examples of commonly used enzymes are herpes simplex virus thymidine kinase (TK) [25], bacterial or yeast-derived cytosine deaminase (CD or yCD) [47,76], bacterial nitroreductase (NTR) [59,77,78], carboxylesterase [79], and carboxypeptidase G2 (CPG2) [80]. Fusion of these proteins, CD-TK [23,24], or improved versions of them were used to arm OAds [60,81]. As suicide genes are non-toxic by themselves and the bystander effect depends on prodrug administration, these armed-OAds could encode the transgene in early or late phase with constitutive promoters or other strategies. Linking their expression to late phase genes may lead to higher amounts of transgene.

Following this rationale, toxins were also used to kill surrounding non-infected cells. For instance, an OAd was armed with melittin, a water-soluble toxic peptide from the bee venom that induces cell apoptosis [82,83]. Toxins fused to antibodies’ single-chains (ScFv), called immunotoxins, were used to target specific cells. An OAd harboring a secreted anti-EGFR-scFv fused to the RNase onconase produced a potent EGFR-dependent bystander killing of tumor cells in vitro and enhanced tumor cell death in vivo [73]. As toxins may have detrimental effects in the virus-producer cell, even with low concentrations, early expression should be avoided to ensure efficient viral replication. For example, linking onconase transcription to late phase with a weak SA was the only arming design leading to viable virus.

Another virocentric approach is to enhance the viral release and spread of OAds. Apoptosis stimulates adenoviral spread through apoptotic bodies [84]. When apoptosis is induced at a late stage of the virus cycle, it does not affect the production of viruses and increases their release from the cell. In consequence, any apoptosis-inducing transgene should be expressed at late phases of the viral cycle. One of the cell death pathways exploited by adenoviruses involves p53. Many cancer cells have non-functional p53 and consequently do no support OAd-induced cell death. Therefore, OAds armed with p53 showed enhanced viral release at the late phase [58,85]. Similarly, other apoptosis inducers have been used to arm OAds such as TNFα [26,27,28,86] or TNF-related apoptosis-inducing ligand (TRAIL). Soluble TRAIL derivatives induce apoptosis independent of p53 status and thus may benefit OAd release in a broad range of tumors. For this reason, sTRAIL has been extensively used in OAds [66,87,88,89,90,91]. Different transgenes that induce late apoptosis have been reported [66,92,93,94,95].

Another strategy to increase the cytotoxicity of OAds is based on RNA interference (RNAi), recently reviewed [50]. Briefly, miRNAs and shRNAs have been inserted to enhance viral replication [96]. In this line, we have recently shown that miR-99b and miR-485 act as enhancers of adenoviral oncolysis by regulating viral transcriptional repressors [97]. Moreover, OAds armed with RNAs (miRNA, shRNA, lncRNA) have been described to silence oncogenes, pro-angiogenic growth factors or apoptosis inhibitors to promote cell death [88,98,99,100,101,102,103,104]. A clear advantage of using RNAs is the absence of packaging limitations, as they tend to be short. However, as with transgenes, their temporal expression must be selected according to its function.

Connective tissue, fibrosis, and extracellular matrix hamper viral spread in the tumor [105]. OAds have been armed to digest the connective tissue. An armed-OAd encoding for relaxin, a peptide hormone able to induce remodeling by degrading collagen and up-regulating matrix metalloproteases, improved the oncolytic potential and tumor spreading in highly metastatic tumor models [106,107]. VCN-01 (Table 1) is an oncolytic adenovirus armed with a soluble version of human hyaluronidase (PH20; *SPAM1*) to degrade hyaluronic acid from the tumor matrix [69,108,109,110,111]. VCN-01 is currently in clinical trials in pancreatic, retinoblastoma, and head and neck cancers. Hyaluronidase has been used also in a chimeric serotype 11/3 OAd. In this backbone, the DNAse I degraded the extracellular DNA, which was also identified as a barrier for interstitial virus spread [112]. Matrix remodeling transgenes don’t affect cell viability allowing early phase expression, but potential toxicity in vivo may still require late and tutor-selective expression.

Fusogenic membrane glycoproteins induce cell-cell fusion and massive syncytia formation. OAds have been armed with measles virus fusogenic membrane glycoprXoteins [113] or gibbon-ape leukemia virus (GALV) envelope glycoprotein [114] significantly enhancing the antitumor efficacy. Also, the formation and subsequent disintegration of the induced syncytia are highly immunostimulatory and have the potential to produce strong systemic bystander effects [115,116]. As adenovirus egress poorly from syncytia, and late phase expression of fusogenic proteins should be considered.

### 4.2. Immunocentric Transgenes

In the last decade, mounting evidence that the immune system can be a powerful ally to cure cancer, mostly fostered by the development of checkpoint inhibitors in oncology, has shifted the strategy in favor of immunocentrism. Oncolytic viruses are an excellent adjuvant for immunotherapy by debulking the tumor, releasing tumor antigens, and providing the appropriate danger signals to inflame tumors and make them susceptible to immunotherapy treatments. As a clear example of this trend, arming OAds with immune-stimulatory cytokines or redirecting immune molecules has emerged as the main approach to prevent cancer spread and recurrence by overcoming the immune tolerance in the tumor. OAds with these transgenes usually contain E3 deletions to provide space and eliminate E3 immune evading functions, and the transgene is expressed early or constitutively. This approach has been widely used, and it is suitable for the most of cases. However, if used systemically such a constitutive expression could lead to toxicity, and it might be appropriate to consider late phase tumor-selective expression.

Granulocyte-macrophage colony-stimulating factor (GM-CSF) stimulates the development, recruitment, activation, and survival of dendritic cells, which are vital for antitumor immunity. ONCOS-102 (Table 1) is an oncolytic virus-based in the Ad5 genome with Ad3 fiber knob and GM-CSF replacing adenoviral E3A. Safety and immunological activity of ONCOS-102 have already been demonstrated in a phase I clinical study for refractory tumors [117,118,119]. CG0070 (Table 1) is also an E3-deleted GM-CSF armed-OAd that in a phase I clinical trial in nonmuscle invasive bladder cancer has shown minor toxicity, viral replication, and high levels of GM-CSF in urine in all patients [120,121].

Interleukin 12 (IL-12) is a heterodimeric cytokine that strongly promotes T-cell helper type 1 (Th1) responses by activating macrophages and dendritic cells. OAds with IL-12 alone or combined with other immune molecules showed antitumor activity even in murine models where human adenoviruses are not able to replicate [122,123,124]. Ad5-yCD/mutKSR39rep-hIL12 (Table 1) is an oncolytic adenovirus combining IL-12 and yeast cytosine deaminase (yCD) which is in a Phase-I clinical trial in pancreatic cancer (NCT03281382). OAds have also been armed with: IL-15 [125,126], IL-21 [127], or IL-24 [128,129,130,131,132,133].

The combination of immunostimulatory cytokines and immune ligands represent the latest trend. Arming with TNFα and IL-2 caused a Th1 enhanced immune response in preclinical models [134] and the OAd TILT-123 advanced to the clinics in melanoma patients receiving an anti-PD1 therapy (Table 1). The LOAd703 (Table 1) is an adenovirus derived from serotype 5 with the fiber from serotype 35, and partial deletion in E3 to express the transgenes trimerized membrane-bound isoleucine zipper (TMZ) TMZ-CD40L and 41BBL under control of a cytomegalovirus (CMV) promoter [135]. It is being tested in clinics as a monotherapy in a variety of cancers or in combination with immunotherapy in malignant melanoma.

Despite the unprecedented efficacy of immune checkpoint inhibitors (ICIs) for certain tumor types, lack of responses in many poorly inflamed (cold) tumors and the systemic autoimmune toxicity have prompted the use of OAds to deliver ICIs. An OAd producing a full-length human monoclonal antibody specific for CTLA4 was published for first time in 2012 [136]. Later, anti-HER2 monoclonal antibody has also been published [137]. NG-350A (Table 1) is an oncolytic adenovirus based on serotypes 11/3 which encodes for a complete anti-CD40. It is currently being tested in clinical trials for metastatic or advanced epithelial tumors administered intratumorally or intravenously.

Armed-OAds evolve in parallel to the immunotherapy landscape. OAds have been designed to produce soluble ligands of immune checkpoints such as OX40L [138], GITRL [139] or a fusion protein sPD1-CD137L [140], and also targeting crucial molecules for the resistance to anti-PD1/PD-L1 and anti-CTL-4 therapies such as TGF-β [141,142]. Moreover, the generation of bispecific T-cell engagers (BiTE) opened a new opportunity for redirecting the anti-viral lymphocytes against tumor-specific antigen. An OAd armed with an EGFR-targeting BiTE showed promising results alone [70] or in combination with CAR-T cell therapy [143]. OAds expressing BiTEs against Epcam [144] or FAP [71] antigens were also reported. NG-641 (Table 1) currently in a clinical trial is armed with 4 transgenes: Interferon alpha (IFNα) to drive dendritic cell priming, CXCL9 and CXCL10 to recruit T-cells and FAP-BiTE to redirect them.

## 5. Conclusions

The best arming strategy depends on the transgene and the desired amount and timing of expression. We have reviewed all published genome insertions sites in oncolytic adenoviruses. Most armed OAds have E3 deletions, which reduces the virus persistence in vivo due to a lower immune-evasion. Thus, we postulated that E3 insertion site could be beneficial for immunocentric transgenes aiming to stimulate the immune system. On the other hand, we suggest that virocentric transgenes or those that need higher expression could be inserted at late sites in non-E3 deleted viruses for longer persistence in vivo, always considering the packaging limits of the virus. In terms of transcriptional control, taking advantage of the endogenous E3 promoters using gene replacements or the major late promoter using splice acceptors allows efficient expression with minimal genomic size increase. In particular the use of the major late promoter allows high expression linked to the virus replication.

## Figures and Tables

**Figure 1 cancers-12-01034-f001:**
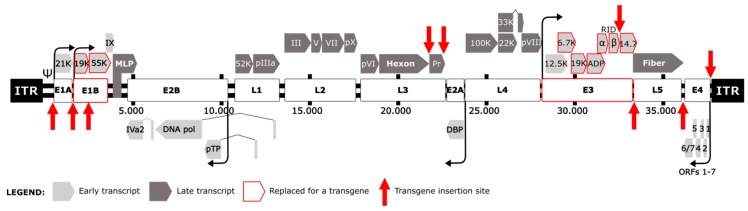
Schematic representation of adenovirus genome. Light grey represents early transcripts (E1A, E1B, E2A, E2B, E3 and E4) and dark grey the late transcripts (L1, L2, L3, L4 and L5) produced by alternative splicing under the major late promoter (MLP). Published insertion sites and replaced adenoviral genes are reported according to the legend.

**Table 1 cancers-12-01034-t001:** Armed oncolytic adenoviruses in clinical trials.

Virus	Backbone	Deletion	Ins. Site	Promoter	Transgene	NCT Identifier
**Ad5yCD/mutKSR39rep-hIL12**	Ad5	E1bΔ55K, ΔE3	E1b55K, E3	CMV	*yCD/TK, hIL-12*	NCT02555397 NCT03281382
**VCN-01**	Ad5	E1aΔ24	After L5	IIIa SA	*PH20*	NCT02045602 NCT03284268 NCT03799744
**CG0070**	Ad5	E1a Δ24 E3Δ19K	Between 6.7 and ADP	End. P	*GM-CSF*	NCT02143804
**ONCOS-102**	Ad5; Ad3 fiber knob	E1aΔ24 E3Δ6.7/19K	E3 6.7/19K	End. P.	*GM-CSF*	NCT03514836 NCT03003676 NCT02963831 NCT02879669
**TILT-123**	Ad5; Ad3 fiber knob	E1aΔ24 ΔE3	E3	End.P.	*TNFα-IRES-* *-IL-2*	NCT04217473
**LoAd703**	Ad5	E1aΔ24; E3Δ6.7/19K	After L5	CMV	*4-1BB CD40L*	NCT03225989 NCT02705196 NCT04123470
**NG-641**	Ad11/3	NA	NA	NA	*IFNα, CXCL9, CXCL10, FAP-BiTE*	NCT04053283
**NG-350A**	Ad11/3	NA	NA	NA	CD40 agonist mAb	NCT03852511

Ins.Site: Insertion Site; CMV: Cytomegalovirus; GM-CSF: Granulocyte Macrophage Colony Stimulating Factor End.P: Endogenous Promoter, FAP-BiTE: Fibroblast Activation Protein - Bispecific Tcell Engager; NA: Information not available.

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
