# Peer review of "Effect of Transgene Location, Transcriptional Control Elements and Transgene Features in Armed Oncolytic Adenoviruses"

_cancers, 2020, doi:10.3390/cancers12041034_

Round 1
Reviewer 1 Report
This review aims to discuss specific strategies used to improve the performance of oncolytic adenovirus treatments with a particular focus on how transgene expression kinetics can be modulated. The introduction is concise and successfully presents the broad concepts in oncolytic viral therapy, though does fail to mention that other oncolytic viruses exist, giving the impression that adenovirus dominates the field, without outlining any key advantage of this virus over others. For the sake of completeness, it would be helpful to frame adenovirus within the OV field and comment on the benefits of adenovirus over other vector systems.
The review in section 2. “Transgene Location” then goes on to introduce the virus and does a good job of briefly explaining how to utilize early and late transcript promoters for maximum effect, as well as explaining draw backs of different strategies. In section 3. “Genetic elements to control transgene expression” the review outlines a number of interesting strategies for using cis/trans acting elements to enhance transgene expression. I would argue this section is the highlight of the review paper and is well written and interesting to read with the various strategies well explained. The authors may want to mention the use of miRNA target sequences here as an additional strategy to regulate and limit transgene expression to the tumor.
Section 4 covers “Codon Usage”, while this part of the review is interesting, in my view this may not warrant an entire section. The broad conclusion of this section might arguably be that ”the virus knows best” how to utilize host aa-tRNAs and while this might be an interesting consideration in designing a transgene cassette, it’s not immediately clear to me how this might be easily leveraged to enhance oncolytic therapies, and no examples of this strategy being utilized in OAd development are given. Additionally, I would be curious to know if aa-tRNA distribution is uniform among different tissues and more importantly among cancers as this would have to be a major consideration if considerable efforts were made to match transgene usage to the target.
The final section (section 5) “Types of Transgenes” is possibly out of place in this review. The review itself largely focuses on specific strategies for improving transgene expression (the title of the review is evidence enough of this). There is a particularly strong focus on ways to use both, endogenous and exogenous cis and trans acting elements to enhance target transgene expression while limiting the size of the transgene cassette. The jump from this into specific transgenes used to enhance oncolytic therapy is jarring and arguably this section warrants an entire review in its own right. This section is probably too short to cover this area in the necessary detail (as well as meandering into checkpoint inhibitor and CAR-T cell research) but is simultaneously mostly technical and does not bring the reader any closer to a conclusion regarding which types of transgenes are most effective for OAd arming. If it is left within this review, some efforts should be made to improve the structure and wording so that the transgene section fits better into the flow of the review, and the title should be changed to reflect this aspect. In Table 1, is not informative or interesting in its current form. Since these are viruses used in clinical trial, some details of the clinical trial should be included, such as the tumor indication(s) investigated and the NCT identifier.
Minor Concerns:
Overall the text should be proofread by a native English speaker. There are many grammatical errors and strange syntax.
Figure 1: A and B are redundant. The same information is shown, except that in B, the sites for transgene insertion are indicated. In my opinion, it would be sufficient to only show B.
Lines 82-84: The final sentence in this paragraph is confusing. I am not clear if the author is referring to deletion of just E3 or E1, E3 and E4, or whether the author is referring to deletion of E3 and then insertion into one of those sites.
Line 187-196: Authors write about 2A short peptide sequences and ribosomal skipping and go on to say that IRES and SA generate better results. They only refer to one paper using the P2A porcine teschovirus-1 2A sequence; however, a number of other 2A sequences exist and have varying ribosomal skipping efficiencies. At line 114 the authors argue that one cannot make a universal statement about cassette orientation, transgene and promoter given all the variables involved. I think it would be reasonable to argue that the same concept might apply here, and it should at least be mentioned that alternative 2A sequences are available and may perform differently.
Figure 2: This figure is rather primitive, and I am not sure it actually adds anything to the manuscript. Should the words that are placed closer to the center indicate genes that have some overlap between the viro- and immune-centric strategies? For example, the word “interleukines” patially lies in the overlapping section; was this intentional? If so, it should be explained. RNAi is in the center, but as far as I can see, there is no mention of RNAi as an immunocentric strategy.
Line 259-260: “Historically, initial arming strategies were more virocentric. They focussed on killing non-infected cells, known as the bystander effect”. This is only partially true. A virocentric approach would also focus strategies to optimize the direct oncolytic effect of the virus, such as evasion from antiviral immune responses, enhancing spread of the virus, maximising CPE in target etc. These types of approaches are discussed and actually account for the majority of the section on virocentric approaches, beginning on line 275. The introduction to this section should be modified to reflect this aspect.
Line 293-294: “micro RNAs miR-99b, miR-485 and shRNA 292 against cullin 4A have been inserted to enhance viral replication.” A short explanation of the mechanism would be helpful.
Reviewer 2 Report
The authors summarized several strategies, such as transgene location, transcriptional control elements, transgene codon usage, and types of transgenes, to genetically improve the therapeutic potential of oncolytic adenoviruses. Although this review was well written, there are some points to address as follows;
- Table 1: The authors demonstrated various types armed oncolytic adenoviruses, which are currently used in clinical trials. Although they demonstrated ClinicalTrials.gov Identifier in the text, there is no ClinicalTrials.gov Identifier in Table 1. The authors should add ClinicalTrials.gov Identifier of each virus in Table 1.
- Table 1: Clinical trials include not only monotherapy but also combination therapy. The authors should add the types of therapy in Table 1.
Reviewer 3 Report
This is very nice review paper. I only have one suggestion. Armed oncolytic adenoviruses are the combination of viruses carrying transgenes as armed and viral cancer selective replication. The introduction only discussed the development of oncolytic approach. It would be more balance if a brief history of armed approach is also included.
The Conclusion is section 6, not 5.
Reviewer 4 Report
This is an excellent review discussing oncolytic adenoviruses and their clinical use. Overall, the review article is well written and does a good job describing the key studies. I only have minor comments to improve the manuscript:
- As the authors mention, oncolytic adenovirus therapy has demonstrated very limited efficacy as a monotherapy. It would be helpful if the authors included a table outlining the key studies where oncolytic adenoviruses were combined with standard of care or targeted agents.
- The clinical trials with immune checkpoint inhibitor therapy are very interesting. I understand that results may not be available for some of the studies, but it would be improve the manuscript if the authors could comment on any preliminary results from these studies, if available.
